# METRIC-OPTIMIZED EXAMPLE WEIGHTS

## ABSTRACT

Real-world machine learning applications often have complex test metrics, and may have training and test data that follow different distributions. We propose addressing these issues by using a weighted loss function with a standard convex loss, but with weights on the training examples that are learned to optimize the test metric of interest on the validation set. These metric-optimized example weights can be learned for any test metric, including black box losses and customized metrics for specific applications. We illustrate the performance of our proposal with public benchmark datasets and real-world applications with domain shift and custom loss functions that balance multiple objectives, impose fairness policies, and are non-convex and non-decomposable.

## 1    INTRODUCTION

In machine learning applications, each training example is usually weighted equally during training. Uniform weighting delivers satisfactory performance when the noise is homoscedastic, training examples follow the same distribution as the test data, and when the training loss matches the test metric. However, these requirements are often violated in real-world applications. For example, the training loss does not always correlate sufficiently with the test metric we care about, such as revenue impact or some fairness metrics. This discrepancy between the loss and metric can lead to inferior test performance (Cortes & Mohri, 2004; Perlich et al., 2003; Davis & Goadrich, 2006).

There are many proposals to address some of these issues, as we review in Section 2, but none of them address all of them at once. In this paper, we propose a framework to learn a weighting function, which estimates training example weights that produce a machine learned model with optimal test metrics. Our proposal, *Metric-Optimized Example Weights (MOEW)*, solves the aforementioned issues simultaneously and is suitable for any standard or customized test metrics. To use MOEW, we need a small set of labeled validation examples that are identically distributed as test examples. By learning a weighting function, MOEW effectively rescales the loss of each training example, and reshapes the total loss function such that its optima better match the optima of the testing metric.

As an illustrative example, Figure 1 shows a simulated non-IID toy dataset and our learned example weighting. The ground truth decision boundary is the diagonal line from the upper left to lower right corners. The feature values in the training and validation/test data follow a beta distribution: specifically, in the training data, $(x_1, x_2) \sim (\beta(2, 1), \beta(2, 1))$, whereas in the validation/test data, $(x_1, x_2) \sim (\beta(1, 2), \beta(1, 2))$. The goal is to maximize precision at 95% recall on the test distribution. With uniform example weights during training, we get 20.8% precision at 95% recall on the test data. With importance weighting, we get 21.8% precision at 95% recall. With MOEW, we obtain 23.2% precision at 95% recall (the Bayes classifier achieves around 25.0% precision at 95% recall). Comparing Figures 1c to 1d, one can see that MOEW learned to upweight the negative training examples, and upweights examples closer to the center.

## 2    RELATED WORK

Our proposal simultaneously addresses three potential issues: non-constant noise levels across training examples, non-identical distributions between training and test examples (a.k.a. covariate shift or domain shift), and training and test objectives mismatch. Several existing methods summarized below address some of these issues, but none of them provide a method to address all of them at once.

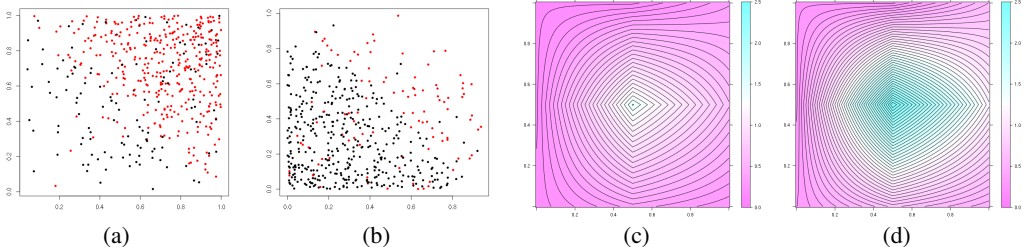

Figure 1: Figures 1a and 1b show the distribution of the training and validation/test data, where black dots represent negative examples and red dots represent positive examples. Figures 1c and 1d show level lines of MOEW for positive and negative examples, respectively.

Maximum likelihood based inference is commonly used to address the issue with non-constant noise levels in the training data. In addition, several common machine learning techniques, such as ensemble and graph-based methods, are found to be robust to binary label noise; see Frenay & Verleysen (2014) for a survey of these methods.

To address the issue of covariate shift, classical approaches include propensity score matching (Rosenbaum & Rubin, 1983; 1985; Lunceford & Davidian, 2004) or importance weighting (Robins et al., 1994; Shimodaira, 2000; Sugiyama et al., 2008; Kanamori et al., 2009). These techniques can also be adapted for model selection tasks (Sugiyama et al., 2007). In addition, Bickel et al. (2009) proposed a discriminative approach of learning under covariate shift.

To address the issue of loss/metric mismatch, researchers have formulated several plug-in approaches and surrogate loss functions. These approaches can be used to optimize some non-decomposable ranking based metrics, such as AUC (Ferri et al., 2002; Yan et al., 2003; Cortes & Mohri, 2004; Freund et al., 2003; Herschtal & Raskutti, 2004; Rudin & Schapire, 2009; Zhao et al., 2011), F score (Joachims, 2005; Jansche, 2005; Dembczynski et al., 2013), and other ranking metrics (Joachims, 2005; Burges et al., 2007; Yue et al., 2007; Eban et al., 2017; Narasimhan et al., 2014; Kar et al., 2014; 2015; Narasimhan et al., 2015). Our work differs in that the test metric can be a black box, and we adapt for it by learning a corresponding weighting function over the examples.

## 3 METRIC ADAPTIVE WEIGHT OPTIMIZATION

We propose learning an example weighting function, trained on validation scores for different example weightings. Our proposal is suitable for any customized metrics.

### 3.1 OVERVIEW

Let $\mathcal{T}$ and $\mathcal{V}$ denote the sets of training and validation examples, respectively. We assume $\mathcal{V}$ is drawn IID with the test set, but that $\mathcal{T}$ and $\mathcal{V}$ may not be IID. Consider a classifier or regressor, $h(x; \theta) \in \mathcal{H}$ for $x \in \mathcal{R}^D$, which is parameterized by $\theta$. Let $y \in \mathcal{R}$ be the label of the data, and $\hat{y} \in \mathcal{R}$ be the predicted score. The optimal $\theta^*$ is obtained by optimizing a weighted loss function $L(\hat{y}, y)$:

$$\theta^*(\alpha) = \arg\min_\theta \sum_{j \in \mathcal{T}} w\left(x_j, y_j; \alpha\right) L\left(h\left(x_j; \theta\right), y_j\right), \tag{1}$$

where $\alpha$ parameterizes the example weighting function $w$. Our goal is to learn an optimal example weighting function $w$ (detailed below).

Let $M_\mathcal{V}(\theta^*(\alpha))$ be the metric of interest evaluated for the model $h$ with parameters $\theta^*(\alpha)$ on the validation dataset $\mathcal{V}$. Without loss of generality, we assume a larger validation metric is more desirable. We propose learning the example weighting function $w(x, y; \alpha^*) \in \mathcal{W}$, such that

$$\alpha^* = \arg\max_\alpha M_\mathcal{V}\left(\theta^*(\alpha)\right). \tag{2}$$

That is, we propose finding the optimal parameters $\alpha^*$ for the example weighting function $w(x, y; \alpha^*)$ such that the model trained with this example weighting function achieves the best validation score.

To simplify the notation, we note that in this paper, the $\theta$ parameters always depend on the weight parameters $\alpha$ and henceforth we simplify the notation $\theta(\alpha)$ to $\theta$.

The metric $M_\mathcal{V}(\theta)$ as a function of $\theta$ and $\alpha$ is likely non-convex and non-differentiable, which makes it hard to directly optimize $M_\mathcal{V}(\theta)$ through e.g., SGD. Instead, we adopt an iterative algorithm to optimize for $\theta^*$ and $\alpha^*$, which is detailed in Algorithm 1. Specifically, we start with a random sample of $K$ choices of weight parameters, $\alpha^0 = \{\alpha_1^0, \ldots, \alpha_K^0\}$. For each of the randomly generated weight parameters $\alpha_l^0$, $1 \le l \le K$, we solve equation 1 to obtain the $K$ corresponding model parameters $\theta_l^0$, and use those to compute the $K$ corresponding validation metrics, $M_\mathcal{V}(\theta_l^0)$. Then, based on the batch of $K$ weight parameters and validation metrics, $\{(\alpha_1^0, M_\mathcal{V}(\theta_1^0)), \ldots, (\alpha_K^0, M_\mathcal{V}(\theta_K^0))\}$, we determine a new set of $K$ weight parameter candidates, $\alpha^1$, and repeat this process for a pre-specified number of iterations, $B$. At the end, we choose the candidate $\alpha$ that produced the best validation metric.

---

**Algorithm 1** Get optimal $\alpha^*$ and $\theta^*$

---

1: **procedure** GET OPTIMAL $\alpha^*$ AND $\theta^*$
2:      $\mathcal{T} \leftarrow$ training data
3:      $\mathcal{V} \leftarrow$ validation data
4:      $B \in \mathbb{N}_+ \leftarrow$ number of batches of weight parameters
5:      $\alpha^0 = \{\alpha_1^0, \ldots, \alpha_K^0\} \leftarrow$ randomly generated initial batch of weight parameters
6:      **for** $i = 0, 1, \ldots, B - 1$ **do**
7:          **parfor** $l \in \{1, \ldots, K\}$ **do**
8:              $\theta_l^i \leftarrow \arg\min_\theta \sum_{j \in \mathcal{T}} w(x_j, y_j; \alpha_l^i) L(h(x_j; \theta), y_j)$
9:          **end parfor**
10:       $s^i \leftarrow \{(\alpha_1^i, M_\mathcal{V}(\theta_1^i)), \ldots, (\alpha_K^i, M_\mathcal{V}(\theta_K^i))\}$
11:       $\alpha^{i+1} \leftarrow$ *GetCandidateAlphas*$(s^0, \ldots, s^i)$
12:      **end for**
13:      $b^*, k^* \leftarrow \arg\max_{b \in \{0, \ldots, B-1\}, k \in \{1, \ldots, K\}} M_\mathcal{V}(\theta_k^b)$
14:      $\alpha^*, \theta^* \leftarrow \alpha_{k^*}^{b^*}, \theta_{k^*}^{b^*}$
15: **end procedure**

---

In the following subsections, we describe the function class $\mathcal{W}$ of the weight model, and the subroutine used in Algorithm 1: *GetCandidateAlphas*.

### 3.2 FUNCTION CLASS FOR THE EXAMPLE WEIGHTING MODEL

Recall that the final optimal $\alpha^*$ is taken to be the best out of $B \times K$ samples of $\alpha$'s. To ensure a sufficient coverage of the weight parameter space for a small number of $B \times K$ validation samples, and to avoid overfitting to the validation data, we found it best to use a function class $\mathcal{W}$ with a small number of parameters.

There are many reasonable strategies for defining $\mathcal{W}$. In this paper, we chose the functional form:

$$w(x, y; \alpha) = c\,\pi(y)\,\sigma\left(z(x, y)^\top \alpha\right), \tag{3}$$

where $c \in \mathcal{R}_+$ is a constant that normalizes the weights over (a batch from) the training set $\mathcal{T}$, and $\sigma(z(x, y)^\top \alpha)$ is a sigmoid transformation of a linear function of a low-dimensional embedding $z$ of $(x, y)$. In the experiments discussed in Section 4, we used the standard importance function $\pi(y) = \frac{p_\mathcal{V}(y)}{p_\mathcal{T}(y)}$, where $p_\mathcal{V}(y)$ and $p_\mathcal{T}(y)$ denote the probability density function of $y$ in the validation and training data, respectively. In practice, $w$ can take many other forms, and $\pi(y)$ can be substituted with any baseline weighting function, which can be considered as an initialization of MOEW.

While there are many possible ways to form a low-dimensional embedding $z$ of $(x, y)$, we choose to use an autoencoder (Hinton & Salakhutdinov, 2006) to form an embedding, as it can be applied to a wide range of problems. Specifically, to train the autoencoder $z(x, y; \psi^*)$, we minimize the weighted sum of the reconstruction loss for $x$ and $y$:

$$\psi^* = \arg\min_\psi \sum_{j \in \mathcal{T}} \lambda\, L_x(z(x_j, y_j; \psi), x_j) + (1 - \lambda)\, L_y(z(x_j, y_j; \psi), y_j), \tag{4}$$

where $L_x$ is an appropriate loss for the feature vector $x$ and $L_y$ is an appropriate loss for the label $y$. The hyperparameter $\lambda$ in equation 4 is used to adjust the relative importance of features and the label

in the embedding: using $\lambda = 0$ is similar to weighting based solely on the value of the label. For all the experiments in this paper we used a fixed $\lambda = 0.5$, but it may need tuning for some problems.

### 3.3 GLOBAL OPTIMIZATION OF WEIGHT FUNCTION PARAMETERS

The validation metric $M_{\mathcal{V}}(\theta^*)$ may have multiple optima as a function of $\alpha$. In order to find the maximum validation score, the sampled candidate $\alpha$'s should achieve two goals. First, $\alpha$ should sufficiently cover the weighting parameter space with a sufficiently fine resolution. In addition, we also need a large number of candidate $\alpha$'s sampled near the most promising local optima. In other words, there is a exploration (spread out candidate $\alpha$'s more evenly) and exploitation (make candidate $\alpha$'s closer to the local optima) trade-off when choosing candidate $\alpha$'s.

One can treat this as a global optimization problem and sample candidate $\alpha$'s with a derivative free optimization algorithm (Conn et al., 2009), such as simulated annealing (van Laarhoven & Aarts, 1987), particle swarm optimization (Kennedy & Eberhart, 1995; Shi & Eberhart, 1998) and differential evolution (Storn & Price, 1997). We chose to base our algorithm on Gaussian process regression (GPR), specifically on the Gaussian Process Upper-Confidence-Bound (GP-UCB) (Auer, 2002; Auer et al., 2002) adapted to batched sampling of parameters (Desautels et al., 2014).

As detailed in Algorithm 2, after getting the $i$-th batch of candidate $\alpha$'s and their corresponding validation metrics $M_{\mathcal{V}}(\theta)$, we build a GPR model $g(\alpha)$ to fit the validation metrics on $\alpha$ for all previous observations. The next batch of candidate $\alpha$'s is then selected sequentially: we first sample an $\alpha_1^{i+1}$ based on the upper bound of the $p\%$ prediction interval of $g(\alpha)$, i.e., $\alpha_1^{i+1} = \arg\max_\alpha Q_{(50+p/2)\%}[g(\alpha)]$. A larger value of hyperparameter $p$ encourages exploration, whereas a smaller value of $p$ encouranges exploitation. After $\alpha_1^{i+1}$ is sampled, we refit a GPR model with an added observation for $\alpha_1^{i+1}$, as if we have observed a validation metric $Q_{(50-q/2)\%}[g(\alpha_1^{i+1})]$, which is the lower bound of the $q\%$ prediction interval of $g(\alpha)$. Hyperparameter $q$ controls how much the refitted GPR model trusts the old GPR model. Using a larger value of $q$ encourages wider exploration within each batch. We then use the refitted GPR model to generate another candidate $\alpha_2^{i+1}$, and continue this process until all the candidate $\alpha$'s in the $(i+1)$-th batch are generated. Note that to ensure convergence, in practice, we usually generate candidate $\alpha$'s within a bounded domain $\mathcal{D}$.

---

**Algorithm 2** Get Candidate $\alpha^{i+1}$

---

1: **procedure** GETCANDIDATEALPHAS
2:     $\mathcal{S} = \{s^0, \ldots, s^i\} \leftarrow$ candidate $\alpha$'s and their corresponding observed validation metrics
3:     $K \in \mathbb{N}_+ \leftarrow$ number of candidates in a batch
4:     $\mathcal{D} \leftarrow$ convex domain in which candidate $\alpha$'s are sampled
5:     $p, q \in [0, 100] \leftarrow$ hyperparameters that control the exploration-exploitation trade-off
6:     **for** $j = 1, 2, \ldots, K$ **do**
7:         $g(\alpha) \leftarrow \text{GPR}(\mathcal{S})$
8:         $\alpha_j^{i+1} \leftarrow \arg\max_{\alpha \in \mathcal{D}} \{Q_{(50+p/2)\%}[g(\alpha)]\}$
9:         $\mathcal{S} \leftarrow \mathcal{S} \cup \{(\alpha_j^{i+1}, Q_{(50-q/2)\%}[g(\alpha_j^{i+1})])\}$
10:     **end for**
11:     $\alpha^{i+1} \leftarrow \{\alpha_1^{i+1}, \ldots, \alpha_K^{i+1}\}$
12: **end procedure**

---

In the experiments presented in Section 4, $p$ and $q$ were set so that the $(50 + p/2)\%$ and $(50 - q/2)\%$ standard Gaussian quantile values equal to $\pm 1$, i.e., $p = q = 68.3$. Figure 2 shows 200 sampled candidate $\alpha$'s in $B = 10$ batches in the example in Section 4.4. It shows that at the beginning of the candidate $\alpha$ sampling process, GPR explores more evenly across the domain $\mathcal{D}$. As the sampling process continues, GPR begins to exploit more heavily near the optimal $\alpha^*$ in the lower right corner.

## 4 EXPERIMENTAL RESULTS

In this section, we illustrate the value of our proposal by comparing it to uniform and importance weightings, i.e., the most common choices in practice, on a diverse set of example problems. For our proposal, we first create a $d$-dimensional embedding of training pairs $\mathcal{T}$ by training an autoencoder

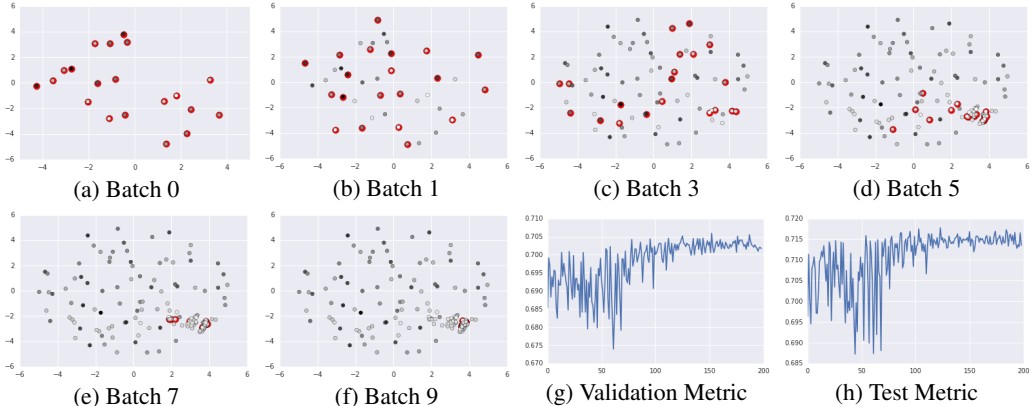

Figure 2: In Figures 2a-2f, red dots show the first two embedding dimensions of candidate $\alpha$'s throughout the $B = 10$ batches GPR sampling process in the example studied in Section 4.4. Figures 2g and 2h show the validation and test metric with the 200 sampled candidate $\alpha$ in 10 batches. It shows that both the validation and test metrics improve as the sampling process progresses.

that has $d$ nodes in one of the hidden layers. We sample $B \times K$ candidate $\alpha$'s ($B$ rounds, $K$ samples each) limited to a $d$-dimensional ball of radius $R$ using a GPR model with an RBF kernel, whose kernel width was conveniently set to be equal to $R$. The noise level of the GP was determined based on the metric noise level of uniform weighting models. For a fair comparison, for uniform and importance weights, we also train the same number of models ($B \times K$) (with fixed example weighting, but random initialization), and pick the model with the best validation metric. Both the autoencoder and the main model were trained for 50k steps using Adam optimizer (Kingma & Ba, 2015) with the default TensorFlow learning rate (0.001). We used squared loss for numeric, hinge loss for binary and cross-entropy loss for multiclass label/features. The learned weights were batch-normalized during training to stabilize the step size. We imposed no regularization on the model parameters, including dropout or early-stopping. To mitigate the randomness in the $\alpha$ sampling and optimization processes, we repeat the whole process 100 times and report the average[1].

Note that the main models in the subsequent examples all have similar structures. The goal is to demonstrate that MOEW could deliver improved performance over baselines regardless of the specific choice of the main model. Here we applied MOEW as a follow-on step after hyperparameter tuning in all of our experiments. In practice, one might get additional gains by tuning MOEW and other model hyperparameters jointly.

The ability to optimize *any testing metric* is a unique benefit of MOEW over other metric-specific methods. In our experiments, we examined MOEW to optimize non-standard metrics. Comparison of our proposal to customized methods for optimizing well-studied testing metrics, such as AUC and F-score (see Section 2), is an open question and a good area of future work.

## 4.1 MOEW PERFORMANCE VERSUS MODEL COMPLEXITY

In this subsection, we empirically examine the performance of MOEW in relation to the complexity of the main model and the weighting model.

### 4.1.1 MAIN MODEL COMPLEXITY

If the model is flexible enough, and the training data clean and sufficient to fit a perfectly accurate model for all parts of the feature space, then the proposed MOEW is not needed. MOEW will be most valuable when the model must take some trade-off, and we can sculpt the loss surface through example weighting in a way that best benefits the test metric. We illustrate this effect by studying the efficacy of MOEW as a function of the model complexity for a fixed size train set. In order to have a controlled experiment, we take a problem that is essentially solved, the MNIST handwritten digit database (LeCun & Cortes, 2010), and train on it with classifiers of varying complexity.

---
[1]The code for experiments in Sections 4.1.1-4.2 will be linked here in the final version.

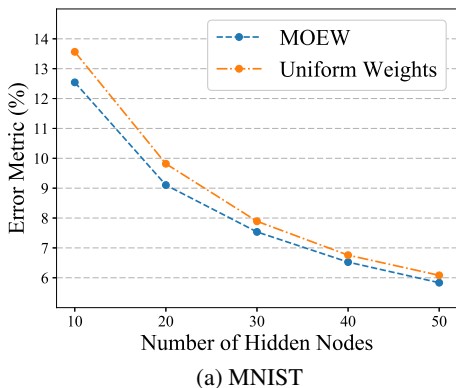 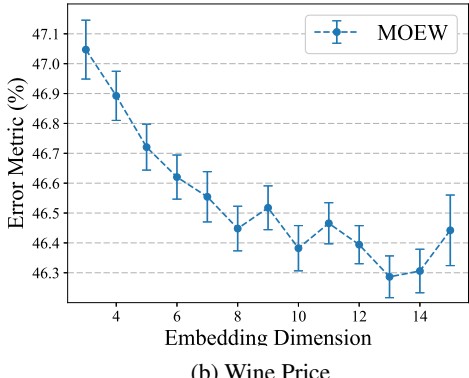

(a) MNIST (b) Wine Price

Figure 3: (a) Error metric on MNIST with varying model complexities. The $95\%$ error margin for any point is less than $0.1\%$. (b) Error metric for the wine price prediction task with varying embedding dimensions. The error metric for the uniform weighting method is $52.00 \pm 0.31$.

We used training/validation/test split of sizes 55k/5k/10k respectively. Features were greyscale pixels $x \in [0,1]^{784}$ and the label $y$ was one-hot encoded in $[0,1]^{10}$. To learn the example weights, we used a 5-dimensional embedding[2], created by training a sigmoid-activation autoencoder network on the $(x,y)$ pair, with $794 \rightarrow 100 \rightarrow 5 \rightarrow 100 \rightarrow 794$ nodes in each layer, and took the activation of the middle layer as the embedding. We used mean squared error for reconstruction of $x$ and cross entropy loss for reconstruction of $y$. The actual classifier was a sigmoid-activation network with $784 \rightarrow h \rightarrow 10$ nodes in each layer. We ran our analysis with varying values for the number of hidden units $h$. We used $B = 60$ batches of $K = 5$ $\alpha$'s in our proposed method and compared against the best-of-300 uniform weighted models[3]. The error metric was taken to be the maximum of the error rates for each digit: $\max_{i \in \{0,\ldots,9\}} \Pr\{\hat{y} \neq i | y = i\}$.

Figure 3a shows the error metric calculated on the testing dataset for increasing model complexities, each averaged over 100 runs. We observe that our proposal clearly outperforms uniform weighting for models of limited complexity. The benefit is smaller for models that are more accurate on the dataset. In most real-world situations, the model complexity is limited either by practical constraints on computational complexity of training/evaluations, or otherwise to avoid overfitting to smaller training datasets. In such cases there might be an inherent trade-off in the learning process (e.g. how much to allocate the capacity of the model to digit 3's that look like 8's v.s. 4's that look like 7's), and we expect our proposed method to apply such trade-off through MOEW.

### 4.1.2 MOEW MODEL COMPLEXITY AND RISK OF OVERFITTING

In this experiment, we study the choice of the embedding dimension for the MOEW algorithm and its risk of overfitting to the validation dataset. We use the Wine reviews dataset from www.kaggle.com/zynicide/wine-reviews. The task is to predict the price of the wine using 39 Boolean characteristic features describing the wine and the quality score (points) for a total of 40 features. We calculate the error in percentage of the correct price, and want the model to have good accuracy across all price ranges. To that end, we set the test metric to be the worst of the errors among 4 quantiles $\{q_i\}$ of the price (thresholds are 17, 25 and 42 dollars): $\max_{i \in \{0,\ldots,3\}} E_{\mathrm{price}(x) \in q_i}[|\hat{y}/y - 1|]$.

We used training/validation/test split of sizes 85k/12k/24k respectively. Because the test metric is normalized by the price, and because the difference in the log space is the log of the ratio $\hat{y}/y$, we apply a log transformation to the label (price) and use mean squared error for the training loss on the log-transformed prices (for all weightings). For MOEW, we illustrate the effect of using a $d$-dimensional embedding for $d \in \{3,\ldots,15\}$, created by training a sigmoid-activation autoencoder network on the $(x,y)$ pair, with $41 \rightarrow 100 \rightarrow d \rightarrow 100 \rightarrow 41$ nodes in each layer, where we took the activation of the middle layer as the embedding. We used mean squared error for the

---

[2]The overall results are similar with embedding dimensions 3 to 8. We chose 5 based on initial experiments.
[3]Smaller values of $K$ result in better exploration, but the runtime can be longer with less parallelism.

Table 1: Average accuracy and fairness violation, together with their 95% error margin with models trained with uniform weighting and the proposed MOEW. We consider two approaches: minimizing fairness violation with identical decision threshold across racial groups (Approach 1), and maximizing accuracy with racial-group-specific thresholds for equal FPR on the training data (Approach 2).

| | Accuracy | | Fairness Violation | |
|---|---|---|---|---|
| | Uniform | MOEW | Uniform | MOEW |
| Approach 1 | $86.82\% \pm 0.07\%$ | $86.84\% \pm 0.12\%$ | $79.23\% \pm 2.35\%$ | $57.74\% \pm 4.64\%$ |
| Approach 2 | $79.12\% \pm 0.12\%$ | $79.95\% \pm 0.17\%$ | $12.26\% \pm 0.30\%$ | $10.06\% \pm 0.54\%$ |

autoencoder reconstruction of $x$ and $y$. The actual regressor was a sigmoid-activation network with $40 \to 20 \to 10 \to 1$ nodes in each layer. We used $B = 10$ batches of $K = 20$ $\alpha$'s in our proposed method and compared against the best-of-200 uniform weighted models.

The model trained with uniform weights had an average test error metric of $52.00 \pm 0.31$. The MOEW method resulted in significantly better test error metrics (i.e. more uniform accuracy across all price ranges). The results with different choices of the embedding dimension, shown in Figure 3b, demonstrate that MOEW can generalize well if the embedding space is small. For larger embedding spaces (say, $d > 10$ in this example), one might need to sample more candidate $\alpha$'s to achieve better convergence and also use a larger validation set to avoid overfitting to the validation set.

## 4.2 COMMUNITY CRIME RATE

In this example, we examine the performance of MOEW with a very small dataset and a complicated testing metric. We use the Communities and Crime dataset from the UCI Machine Learning Repository (Dheeru & Karra Taniskidou, 2017), which contains the violent crime rate of 994/500/500 communities for training/evaluation/testing. The goal is to predict whether a community has violent crime rate per 100k popuation above 0.28.

In addition to obtaining an accurate classifier, we also aim to improve its *fairness*. To this end, we divided the communities into 4 groups based on the quantiles of the percentage of white population in each community (thresholds are 63%, 84% and 94%). We seek a classifier with good accuracy, and at the same time have similar false positive rates (FPR) across racial groups. Therefore, we evaluate classifiers based on two metrics: overall accuracy across all communities and the difference between the highest and lowest FPR across four racial groups (fairness violation).

We used a linear classifier with 95 features, including the percentage of African American, Asian, Hispanic and white population. For MOEW, those 95 features plus the binary label were projected onto a 4-dimensional space using an autoencoder with $96 \to 10 \to 4 \to 10 \to 96$ nodes. We sampled candidate $\alpha$'s in 10 batches of size 5, and compared our proposal against the best-of-50 uniform weighted models.

In practice, there is usually a trade-off between accuracy and fairness of classifiers (see, e.g., Hardt et al. (2016); Goh et al. (2016)). To explore this trade-off, we considered two approaches: with the first (vanilla) approach, we used MEOW to minimize the difference between the highest and lowest FPR across the 4 groups (i.e., minimize fairness violation), with identical decision thresholds for all groups. With the second approach, after training the model, we set a different decision threshold for each racial group to achieve the same FPR on the training data while maintaining the same overall coverage (post-shifting) (Hardt et al., 2016), and used MOEW to maximize the final accuracy.

The results are summarized in Table 1. With the first approach, MOEW reduces the fairness violation by over 20% and yet achieves the same overall accuracy compared to uniform weighting. With the second approach, MOEW improves both accuracy and reduces the fairness violation.

## 4.3 SPAM BLOCKING

For this problem from a large internet services company, the goal is to classify whether a result is spam, and this decision affects whether the result provider receives ads revenue from the company. Thus, it is more important to block more expensive spam results, but it is also important not to block any results that are not spam, especially results with many impressions. We use a simplified

Table 2: Average test metric and 95% error margin with models trained with uniform example weights, importance weighting and the proposed MOEW. Larger test metric is better.

|  | Uniform | Importance | MOEW |
|---|---|---|---|
| Spam Blocking | $1.000 \pm 0.040$ | $1.210 \pm 0.063$ | $1.849 \pm 0.133$ |
| Web Page Quality | $0.7113 \pm 0.0004$ | $0.7113 \pm 0.0004$ | $0.7176 \pm 0.0004$ |

test metric that captures these different goals (the actual metric is more complex and proprietary). Specifically, for each method we set the classifier decision threshold so that 5% of the test set is labelled as spam. We then sum the costs saved by blocking correctly identified spam results and divide it by the total number of blocked impressions of incorrectly-identified spam results.

The datasets contain 12 features. The 180k training dataset is 25% spam, and is not IID with the validation/test datasets, which are IID and have 10k/30k examples respectively with 5% spam.

We trained an autoencoder on the 12 features plus label, with layers of $13 \rightarrow 100 \rightarrow 3 \rightarrow 100 \rightarrow 13$ nodes, and used the middle layer as a 3-dimensional embedding. For each weighting method, we built a sigmoid-activation network classifier with architecture $12 \rightarrow 20 \rightarrow 10 \rightarrow 1$. Candidate $\alpha$'s were sampled $K = 20$ at a time in $B = 10$ rounds of sampling.

Table 2 compares the two non-uniform example weighting methods to the uniform test metric, where we have normalized the reported scores so that the average uniformly weighted test metric is 1.0. Our proposed method clearly outperforms both uniform and importance weightings.

## 4.4 WEB PAGE QUALITY

This binary classifier example is from a large internet services company. The goal is to identify high quality webpages, and the classifier threshold is set such that 40% of examples are labelled as high quality. The company performed several rounds of human ratings of example web pages. The label generated in the early rounds was a binary label (high/low quality). Then in later rounds, the human raters provided a small number of examples with a finer-grained label, scoring the quality in $[0, 1]$. The test metric for this problem is the average numeric score of the positively classified test examples.

On the 62k training web pages that have the binary labels, we trained a six-feature sigmoid-activation classifier with $6 \rightarrow 20 \rightarrow 10 \rightarrow 1$ nodes. To learn the proposed MOEW, we trained an autoencoder that mapped the six features plus label onto a 3-dimensional space: $7 \rightarrow 100 \rightarrow 3 \rightarrow 100 \rightarrow 7$. We sampled $K = 20$ candidate $\alpha$'s for $B = 10$ rounds.

The validation data and test data each have 10k web pages labeled with a fine-grained score in $[0, 1]$. The datasets are not IID: the training data is the oldest data, the test data is the newest data, with validation data in-between. We summarize the average quality score of 40% selected web pages in the test data in Table 2, together with the 95% error margins. Note that in this example, importance weighting results in uniform weighting.

## 5 CONCLUSIONS

In this paper, we proposed learning example weights to sculpt a standard convex loss function into one that better optimizes the test metric for the test distribution. We demonstrated substantial benefits on public benchmark datasets and real-world applications, for problems with non-IID data, heterogenous noise, and custom metrics that incorporated multiple objectives and fairness metrics.

To limit the need for validation data and re-trainings, we tried to minimize the free parameters of the example weighting function. To that end, we used the low-dimensional embedding of $(x, y)$ provided by an autoencoder, but hypothesize that a discriminatively-trained embedding could more optimal.

We hypothesize that the MEOW could also be useful for other purposes. For example, they may be useful for guiding active sampling, suggesting one should sample more examples in feature regions with high training weights. And we hypothesize one could downsample areas of low-weight to reduce the size of training data for faster training and iterations, without sacrificing test performance.

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
