# OpenReview forum: "Metric-Optimized Example Weights"
_ICLR.cc/2019/Conference_

### Official Review · AnonReviewer2 · 2018-11-02
**This paper motivates using a nonlinear function based on an autoencoded representation to derive training sample weights to optimize any test metric. It seems like a significant contribution, particularly for analyzing datasets where training and validation data are known to be drawn from different distributions.**

**Rating:** 7
**Confidence:** 3

**Review:**

This proposal learns training data weights that optimize any given test metric.
They do this by learning both a weighting function and a classifier. They
iteratively train an ensemble of K {weighting_k,classifier} pairs, and
selecting the pair that presents the best best metric-of-interest value over
the test set (over all iterations) as the final output.

The paper provides a large set of references that I found useful, and is
clearly written.

Each training iteration of their MOEW algorithm first optimizes classifiers
over all training examples, for each of the K sample weightings.  Each
weighting gives rise to one converged classifier, whose metric-of-interest is
evaluated on the validation set.  Given K sets of weighting functions and their
metric-of-interest values, new parameters for the weighting functions are
generated.

The weighting function choice is a simple function, based on a linear transform
of autoencoder features, a normalization factor, and factor to account for
differing label frequencies in training and test data.  The low-dimensional
auto-encoding of {data,label} pairs is trained once, using the training data.
The weighting function parameters are a linear transformation matrix. They
also have a sigmoid non-linearity in their weighting function.

If I understand correctly, their approach seems to generalize the importance
sampling methods that they reference by using a nonlinear combination of autoencoder
features.  They reference importance sampling methods that, for example, use
Gaussian kernel basis functions to model training/test example densities.
While they do provide extensive references to previous approaches, I would have
enjoyed a clearer explanation of main differences between their weighting
function and ones that have previously been used in importance sampling.

Their main contribution seems to be the procedure (Alg. 2) used to updated the
weighting function parameters.  They first fit a model that predicts
metric-of-interest values given all previous weighting function parameters used
in the algorithm.  Then they use this model to generate the next set of
weighting function parameters.  Their method adapts a Gaussian Process
Upper-Confidence Bound to batch-wise processing.  Within-batch and between
batch exploration of the weighting function parameters is controlled by two
parameters, which they held fixed at values corresponding to +/-1 for a normal
distribution.

Only the validation set need be iid with the test set, so their method seems
quite general.

Their MNIST results use very simple networks to show that learning weighting
functions has the most benefit when classifier networks are severely
under-parameterized.  Their wine price example addresses the choice of
embedding dimension, and they found their error metric decreased from 52% with
uniform weighting, to around 46% as they approached ~10 dimensions in the
autoencoded {data,label} representation of the training data.  They then use a
small crime dataset with a complicated test metric measuring *fairness*, based
on dividing the dataset into 4 quantiles based on white population. Using
MOEW they could improve the fairness metric with little effect on the accuracy
metric.  They could also preset thresholds to achieve very good fairness on the
training data and use MOEW to maximize the accuracy metric, which resulting
in improving both fairness and accuracy compared to uniform sample weighting.

For spam blocking and web page quality, I would like to see a little more
interpretation of their results. They again show improvements using MOEW to
provide weights for training data.  This time, Table 2 presents a comparison
with importance sampling, but the methodology they used for importance sampling
is not well described.  I'd like to understand where this improvement came
from. What model/procedure was used for importance weighting? Gaussian rbfs?
Do the authors attribute the dramatic improvement for MOEW for Spam Blocking
simply to having a more flexible sample distribution function?  Or is it mainly
due to their adapting to the test metric?  Then for Web Page Quality, would it
be correct to conclude that the old and new web pages (training vs
validation/test) are actually fairly similary distributed?

What values of B and K are reasonable values in Alg. 1?  When applied to larger
problems, where do the authors feel the bottlenecks in Algs 1 and 2 will lie?
Do the authors find that retuning the classifier for different weighted samples
to take a lot of time?

Pros:
 - tests on several datasets
 - seems fairly generally applicable.

Cons:
 - To reproduce, I guess I'd need to adapt existing GP-UCB code from python or
   C++, and I'm not sure how easy this would be. Releasing code to reproduce
   the results might be nice.
 - Better understanding of differences wrt. typical Importance Sampling methods
   would nice.

---

### Official Review · AnonReviewer1 · 2018-11-04
**Interesting problem. I think more discussion and experiments to understand better potential overfitting issues are needed here.  I'm intrigued that the approach works as well as it does on the examples given, so there may be something interesting here.**

**Rating:** 4
**Confidence:** 4

**Review:**

Pros:
- Addresses several interesting and important problems all at once: covariate shift, concept drift, mismatch between training loss and test loss.
- Fairly simple and elegant solution.
- Multiple examples of the method working.
- Clearly written

Cons:
- Examples don't feel like full-fledged machine learning examples, where you  tune your learning algorithm on the validation set, as well as the example weights (their approach).
- Needs discussion of potential overfitting issues (see comments below).

Comments:
- I think one area that is underexplored in this paper is overfitting.  One issue is overfitting the validation data by having more complicated weight functions. For example, in Figure 3(b), as the embedding dimension goes beyond 14, it seems like the error metric gets worse.  Would be interesting to see a plot of the validation error metric alongside this test error metric.  Also, what happens when we use even larger embedding dimensions -- that should clarify whether this is an overfitting situation, or just a random chance fluctuation.
- In machine learning contexts, it's standard to try many different ML methods (or at least network architectures) with various hyperparameter settings and regularization methods, yet this isn't discussed at all in the paper.  Would you use the search for alpha as an inner loop in your model search and hyperparameter selection process?  I'd expect there could be additional issues with overfitting the validation set as you used more complicated models.  I think not discussing or investigating this makes the examples feel a little bit more like toys.  I think tuning your learning algorithm settings on a validation set is pretty intrinsic to machine learning approaches.
- Relatedly, you say "we impose no regularization on the model parameters"... does this include things like early stopping, dropout, or other things that are used to prevent overfitting?  This seems just part of the "no hyperparameter tuning" setting of the paper.
- In the introdution you say "MOEW . . . reshapes the total loss function to better match the testing metric.  This is similar to the idea of basis expansion, where we approximate the metric function using a linear combination of per example loss functions".  You make a similar statement in the conclusion. This is an interesting idea, but it doesn't seem to represent what you're doing. This explanation suggests that you are fitting \alpha's so that the objective function value approximates the validation metric for each theta.  But that's not what you're doing, right?
- In 3.2, you say that c "is a constant that normalizes the weights over a (batch from) the training set T".  Why would you renormalize per batch?  This seems to potentially negate the effect of the reweighting, especially for small batches.
- It might have been interesting to see if there was any significant differences between hinge loss and cross-entropy loss for the binary case.
- In the MNIST experiment, am I correctly understanding that the difference between the 300 uniform weighted models you tried was the random initialization of the weights? Was there a lot of variation in performance among these trials?  Folk wisdom makes me think there would not be large performance differences.

---

### Official Review · AnonReviewer3 · 2018-11-06
**good initiative, not mature enough**

**Rating:** 4
**Confidence:** 4

**Review:**

The authors propose to optimize a black-box (validation/test) metric by learning to re-weight the training examples. The weights are calculated from a linear model on an auto-encoder-computed embedding, and the parameters of the linear model is found by an Gaussian-Process-Regression-(UCB)-guided global optimization procedure. Experimental results demonstrate that the learnt weights outputs uniform weights.

The paper is well-written with clear motivations. Nevertheless, the paper is not mature due to the following reasons:

(1) The alternatives are not carefully compared/discussed for the key components of the proposed framework.
   (1a) Why need the linear model? What if the weights are not calculated from the linear model parameters, but optimized by GPUCB directly?
   (1b) Why need GPUCB? What if we just do random search or standard simulated annealing for global optimization?
   (1c) Following (1a) and (1b), what if the weights are optimized directly through random search?
   (1d) What if, in the case of MNIST, class weights are used instead of example weights
   (1e) How sensitive is the proposed framework in terms of hyperparameters like p and q, and perhaps other GP parameters?

(2) No comparison on the standard-but-challenging metrics like F1. The authors state in the experiments that it is not the focus of the work, but I do believe that the comparison is meaningful to help understand whether the proposed framework is close to the state-of-the-art in those standard metrics. Otherwise the baseline (uniform weights) is arguably just too weak.

(3) Is the framework just overfitting the validation data set by reusing it to evaluate multiple times? Are there overfitting behavior observed during the reuses?

---

### Meta-Review · Area_Chair1 · 2018-12-15
**Not ready for publication at ICLR**

**Confidence:** 5
**Recommendation:** Reject

**Metareview:**

While there was some support for the ideas presented, the majority of reviewers did not think this paper was ready for publication at ICLR. In particular the experiments need more work, including the protocol for validation, and attention to overfitting.